# Nile Tilapia (*Oreochromis niloticus* Linnaeus, 1758) Invasion Caused Trophic Structure Disruptions of Fish Communities in the South China River—Pearl River

**DOI:** 10.3390/biology11111665

**Published:** 2022-11-15

**Authors:** Fangmin Shuai, Jie Li

**Affiliations:** Pearl River Fisheries Research Institute, Chinese Academy of Fishery Sciences, Guangzhou 510380, China

**Keywords:** invasion, isotope structure, trophic status, isotopic diversity, isotopic niche

## Abstract

**Simple Summary:**

Biological invasions have become an important part of global change and result in devastating ecological and economic impacts worldwide. Nile tilapia (*Oreochromis niloticus* Linnaeus, 1758) have been introduced to at least 100 countries for aquaculture, while it is currently recognized as one of the most dangerous invasive species in the tropical and subtropical regions of the world. This study analyzed how Nile tilapia invasion disrupts the trophic structure of native species. The results showed that Nile tilapia invasion reduced the trophic status, shortened the food chain, and affected the isotopic diversity of native fish species. This study provided clear evidence that invasive Nile tilapia could destroy recipient ecosystem stability by disrupting the trophic structure and food chains of native communities.

**Abstract:**

Widespread introductions of non-native species, including aquaculture and ornamental species, threaten biodiversity and ecosystem functioning by modifying the trophic structure of communities. In this study, we quantified the multiple facets of trophic disruption in freshwater communities invaded by Nile tilapia, by comparing uninvaded and invaded rivers downstream of the Pearl River, China. Nile tilapia invasion reduced the trophic status of native fish species by forcing native herbivores and planktivores to seek new food sources. The food chain was also shortened by decreasing the trophic levels of native invertivores, omnivores, and piscivores, while the total isotopic niche area (TA) of native invertivores, omnivores, piscivores, and planktivores species also decreased. Simultaneously, Nile tilapia invasion affected the isotopic diversity of the fish community. Decreasing isotopic richness (IRic), isotopic evenness (IEve), and increasing isotopic uniqueness (IUni) indicated that Nile tilapia had a high trophic niche overlap with native species and competed with native species for food resources, and even caused the compression of the trophic niche of native species. Understanding the process described in this study is essential to conserve the stability of freshwater ecosystems, and improve the control strategy of alien aquatic organisms in south China.

## 1. Introduction

Biological invasions have become an important part of global change and result in devastating ecological and economic impacts worldwide [1]. Numerous studies have documented the decline of native species following the establishment of invasive species [2,3,4]. For instance, the establishment of Round Goby (*Neogobius melanostomus* Pallas, 1814) has been associated with decreases in the occurrence of native fish such as Mottled Sculpin (*Cottus bairdii* Girard, 1850) in the Laurentian Great Lakes [5], three-spined Stickle-back (*Gasterosteus aculeatus* Linnaeus, 1758) in the Gulf of Gdansk, and the protected River Bullhead (*Cottus perifretum* Freyhof, Kottelat & Nolte, 2005) in the River Meuse, Netherlands [6]. The widespread introduction of non-native fish can modify recipient communities through notable changes in biotic interactions between species, such as predation and competition [7,8]. This damage has prompted recent efforts to explore the impact mechanisms of species invasion on ecosystems and prioritize them for ecological management [8,9,10].

More importantly, biological invasions are considered to be one of the most serious threats to ecosystem functioning [11,12,13]. Food webs are the most important and basic network linking organisms in a community, and are also a key feature of ecosystem stability [14]. Therefore, the impact of non-native species on the food web will certainly disrupt the nutritional structure of natural communities and ultimately destroy the ecological function of the ecosystem [15,16]. However, the mechanisms by which invasive species disrupt the food web and trophic structure of native species is still not clear [11]. This is mainly because it is difficult to quantify the impact of species invasion on the nutritional structure of a community. In addition, the structures of food webs are often very complex, and the feeding habits of species in a community are variable, and animals will adapt their diets to overcome increasing food competition with invasive species [17]. As invasive species typically have broad diets, they can interact with a wide variety of prey species at multiple trophic levels [18].

Gut content analysis has been traditionally used to evaluate trophic relationships among organisms. However, this approach only provides a snapshot of predator–prey interactions, and has some shortcomings such as a high rate of empty stomachs, being labor intensive, and extremely time consuming [19]. Stable isotope technology is continuously improving, and is based on the relationship between the isotopic composition of consumers and their diet, providing a comprehensive tool to quantitatively detect interactions among organisms [15,19,20]. Using this technology, ecologists can quantitatively analyze the various impacts of invasive species on the trophic structure of native species [21,22,23]. For instance, using stable isotope technology, Koel et al. [24] explained how the invasion of lake trout in Yellowstone Lake induced within and across ecosystem effects by forcing native cutthroat trout to shift their diets to incorporate a higher proportion of amphipods, thus increasing the trophic level of native cutthroat trout. The invasion of lake trout altered plankton assemblages and nutrient transport to tributary streams, forcing grizzly bears and black bears to seek alternative food sources [24].

Freshwater ecosystems are widely considered to be one of the most important ecosystems supporting human survival. They are rich in biodiversity and provide irreplaceable ecosystem services to humans, including essential drinking water and fish products [25]. However, anthropogenic pressures, including overfishing, pollution, and the introduction of non-native species, have resulted in a sharp decline in freshwater ecosystem functioning [26]. Freshwater ecosystems can easily be destroyed and are difficult to restore. They are also regarded as one of the most endangered ecosystems [27,28,29]. Therefore, freshwater ecosystems should be of high priority for management and conservation. Especially under the context of global change, the study of freshwater ecosystems is of great significance for maintaining human health and harmony [27].

South China is warm and rainy, with an average annual temperature of 23 °C and an average annual precipitation of 1800 mm. This region supports high levels of aquatic biological resources and endemism. It is a hot spot for global biodiversity research [30]. The warm climate and abundant rainfall means that southern China has an extensive aquaculture industry and has the highest frequency of non-native fish species in the world [31]. The most frequently introduced freshwater fish in south China is Nile tilapia. Nile tilapia are native to Africa and have been introduced to at least 100 countries for aquaculture due to their fast growth, disease resistance, environmental tolerance, good meat quality, and high yield [32]. At present, it is one of the most important freshwater aquaculture species in the world [33]. However, viable wild populations of Nile tilapia are now established in most tropical and subtropical waters globally [34], including Lake Victoria, the River Limpopo system (Africa), water bodies in Pennsylvania and Mississippi (USA), and the Pearl River system in China [35,36]. Nile tilapia was first introduced into Guangdong Province in southern China for aquaculture in 1957. Since then, a large number of Nile tilapia have escaped into natural water bodies. Known for their capacity to cause a series of ecological and environmental problems, such as trophic cascades, changes in water quality, habitat degradation, and alterations in ecosystem function [35,37,38], Nile tilapia are currently recognized as one of the most dangerous invasive species in the tropical and subtropical regions of the world [39].

Although a number of studies have examined the adverse effects of Nile tilapia invasion on aquatic ecosystems [37,40], details on the trophic status of native species are generally lacking, with the exception of a few studies [41]. The purpose of the current study is to reveal how Nile tilapia invasion disrupts the trophic structure of native species. Specifically, we quantified how trophic status and isotopic diversity of native fish changed as a result of Nile tilapia invasion, because the isotopic niche corresponds to a portion of the ecological niche and isotopic diversity represents patterns of resource and habitat use of organisms [22,42]. We divided native species into six categories (piscivores, invertivores, omnivores, detritivores, herbivores, and planktivores) based on their feeding habits, and further analyzed the effects of Nile tilapia invasion on different fish trophic niches. Empirical examples of interactions between non-native and native species in the wild are scarce because studies often lack pre-invasion data, thereby preventing before–after study designs. In this study, the target Dongjiang River has a serious Nile tilapia invasion as a result of the aquaculture industry. The reference parallel Beijiang River has a relatively small Nile tilapia population due to an underdeveloped aquaculture industry. It is not invaded according to long-term fishery resources monitoring, and was used for comparison in the absence of pre-invasion data. Understanding the process described in this study is essential to conserve the stability of freshwater ecosystems, and improve the control strategy of alien aquatic organisms in south China.

## 2. Materials and Methods

### 2.1. Study Area

The current study was conducted in the Dongjiang River and Beijiang River in southern China. Four sampling sites in the invaded Dongjiang River and four in the non-invaded Beijiang River were established (Figure 1, Table 1).

### 2.2. Data Collection

Fishing is prohibited in the south of China from March to June every year, including the Dongjiang and Beijiang River basins. To avoid differences caused by seasons, fish samples were collected twice a month in July, August, September, and October at each sampling site from 2013 to 2021. Catch samples were collected over one to three successive days at each site, according to favorable weather conditions. A combination of various fishing nets was used to overcome selectivity effects of a single net on fish species. The fishing nets included a set of gillnets (length: 10 m, height: 2.5 m; mesh size: 20 mm), fishing hooks (length: 20 m, hooks: 50), and lobster pots (length: 15 m, radius: 18 cm). Sampling was performed using the same protocol in each year. Sampling started in the afternoon (approx. 18:00 pm) and lasted 12 h for a whole night for all the nets. All sampled individuals were identified to species level and measured (total length, mm; wet weight, g) immediately on site. For fish isotope sample collection, the white muscles were dissected from the upper side of the body, and put into a 5 mL centrifuge tube. Isotopic muscle samples were only collected from adult individuals to reduce any possible confounding effects of life stage on isotopic values [43]. All muscle samples were dried to a constant weight at 60 °C and ground into powder in the laboratory. Each sample weighed at least 0.5 mg and had at least six replicates.

### 2.3. Stable Isotope Metrics

C and N isotope analysis was carried out on a Finnigan Delta V Advantage Isotope Ratio Mass Spectrometer (IRMS, Thermo Fisher Scientific, Inc., Waltham, MA, USA) and a Flash 2000 HT Elemental Analyzer (Thermo Fisher Scientific, Inc., Waltham, MA, USA). The δ^13^C range represents the broadness of resources used in a community, and the δ^15^N range represents the trophic levels (i.e., food chain length) within a community. The δ^13^C range and δ^15^N range of native species in each river were first calculated as the differences between maximal and minimal values for δ^13^C and δ^15^N to compare the difference in resource use and food chain length between the invaded and reference rivers [8,20]. Then, trophic niche size was calculated using two complementary methods. The total isotopic niche size (TA) was quantified as the total area within the convex hull shaping the community in the two-dimensional δ^13^C-δ^15^N space [20]. The core isotopic niche (SEA) was calculated as the standard ellipse area based on Bayesian statistics, which reflected the core of the trophic niche. SEA is less sensitive to extreme values than TA and provides complementary information about the isotopic niche of a community [15].

### 2.4. Isotopic Diversity Indices

Four complementary isotopic diversity indices were selected in the current study to quantify the impacts of Nile tilapia invasion on the isotopic diversity of native species. These indices are: isotopic richness (IRic), represents the amount of isotopic space filled by a group of organisms; isotopic evenness (IEve), describes the regularity in the distribution of organisms and of their weight in the stable isotope space; isotopic divergence (IDiv), describes the distribution of organism importance within the border context of the stable isotope space; and isotopic uniqueness (IUni), is the weighted-average distance divided by the maximal distance between two nearest neighbors, namely the inverse of the average isotopic redundancy. These indexes were calculated following Cucherousset and Villéger [22].

### 2.5. Statistical Analyses

A linear mixed effect model (isotopic metric ~ number of native piscivores + number of native invertivores + number of native omnivores + number of native detritivores + number of native herbivores + number of native planktivores + number of Nile tilapia population) was carried out to test the effects of Nile tilapia on the δ^13^C and δ^15^N ranges and on the TA of native species. Mixed effect models were computed using the nlme package in R 3.3.1 [44]. An independent sample t-test was used to analyze the differences between variables of the invaded river and the reference river. The abundance data were lg(x + 1) transformed before analysis to conform to the normal distribution. All analyses were conducted using R Statistical Software version 3.3.1 [44]. Variables were considered statistically significant at *p* < 0.05.

## 3. Results

### 3.1. Fish Community Structure

A total of 13 piscivores species, 17 invertivores species, 25 omnivores species, five detritivores species, 10 herbivores species, and three planktivore species were sampled during the present study in the invaded Dongjiang River. Of these, 64 were native and 9 were non–native species. Of the nine non–native species, Nile tilapia were the most abundant, accounting for 13.21% of all individuals in the Dongjiang River (Table 2). The abundance of the other non-native species was very low. A total of 18 piscivores species, 13 invertivores species, 24 omnivore species, 5 detritivores species, 13 herbivore species, and 4 planktivores species were sampled in the reference Beijiang River. Of these,

Seventy were native and seven were non-native species. The abundance of all the non-native species was very low, and Nile tilapia abundance accounted for 3.65% of all individuals (Table 2). Compared to the reference Beijiang River, the invaded Dongjiang River had fewer piscivores, and omnivores, but more herbivores (Figure 2).

### 3.2. Relationship between Nile tilapia Invasion and Trophic Structure

In the invaded Dongjiang River, the δ^13^C range significantly increased with the number of native piscivores (*p* = 0.019) and the number Nile tilapia (*p* = 0.030) (Table 3). The δ^15^N range significantly increased with the number of native piscivores (*p* = 0.008) and planktivores (*p* = 0.0031), but significantly decreased with the number of Nile tilapia (*p* = 0.025; Table 3). The other feeding habits had no significant effect on δ^13^C range or δ^15^N range. In the reference Beijiang River, the δ^13^C range significantly increased with the number of native detritivorous fish (*p* = 0.042) and piscivores (*p* = 0.037) (Table 3). The δ^15^N range significantly increased with the number of native piscivore fish (*p* = 0.036; Table 3) alone. The other feeding habits and Nile tilapia had no significant effects on δ^13^C and δ^15^N range.

In the invaded Dongjiang River, herbivores (*t* = −2.28, *p* = 0.031) and planktivores (*t* = −2.83, *p* = 0.038) had a longer δ^13^C range than the reference Beijiang River, while piscivores (*t* = 2.66, *p* = 0.014) had a shorter δ^13^C range than in the reference Beijiang River (Figure 3a). In the invaded Dongjiang River, invertivores (*t* = −9.36, *p* = 0.003), omnivores (*t* = −3.28, *p* = 0.002), and piscivores (*t* = −4.01, *p* < 0.001) had a shorter δ^15^N range than the reference Beijiang River. The other feeding habits had no significant difference on δ^13^C and δ^15^N range (Figure 3b). On the whole, the δ^13^C range in the invaded Dongjiang River was longer than that in the reference Beijiang River (Figure 3c), while the δ^15^N range was shorter (Figure 3d).

Further analysis revealed that invertivores (*t* = −2.70, *p* = 0.04), omnivores (*t* = −2.86, *p* = 0.032), piscivores (*t* = −1.87, *p* = 0.044), and planktivores (*t* = −2.69, *p* = 0.018) in the invaded Dongjiang River had a lower TA and a lower SEA than the reference Beijiang River, while the detritivores (*t* = 1.59, *p* = 0.027) in the invaded Dongjiang River had a higher TA and a higher SEA than the reference Beijiang River (Figure 4a,b). In general, the invaded Dongjiang River had a smaller TA (*t* = −2.67, *p* = 0.011) and smaller SEA (*t* = −2.59, *p* = 0.015) than the reference Beijiang River (Figure 4c).

In the sampled fish communities, IDiv showed no obvious difference between rivers. IEve (*t* = −1.66, *p* = 0.048) and IRic (*t* = −2.77, *p* = 0.032) in the invaded Dongjiang River were significantly lower than the reference Beijiang River (Figure 5). The IUni (*t* = 2.19, *p* = 0.034) in the invaded Dongjiang River was significantly higher than the reference Beijiang River (Figure 5).

## 4. Discussion

Freshwater ecosystems are considered to be among the most altered ecosystems in the world, especially because of the widespread introduction of non-native fish [1]. Introduced fish species have been shown to alter existing biological interactions among native species [3] and modify the overall trophic structure of recipient communities. For instance, introduced predators can increase food chain length [47]. In the current study, we provided empirical evidence that Nile tilapia invasion destabilized the isotopic structure of native fish species through a stepwise process of trophic disruptions.

In the invaded Dongjiang River, the δ^13^C range significantly increased. This indicated that the Nile tilapia invasion produced food competition with native fish, forcing native fish species to seek new food sources, as an increased δ^1^3C range represents wider food resources in a community. Specifically, Nile tilapia invasion forced the low trophic status native herbivores and planktivores to seek new food sources (with a longer δ^13^C range than that in the reference Beijiang River). For native fish, these disruptions increased competition, which promotes the dietary generalists at the expense of specialists [48].

Although differences in δ^13^C ranges may be driven by differential contributions of the basal resources fueling a community [49], the results of the current study showed that the Nile tilapia invasion could modify the basal food resources in recipient communities. Simultaneously, Nile tilapia invasion reduced the food resources available for native piscivores (a shorter δ^13^C range than the reference Beijiang River). There has been evidence that the invasion of Nile tilapia in the Pearl River significantly reduced the abundance of native fish species [38], including the dominant native mud carp (*Cirrhinus molitorella* Valenciennes, 1844), black amur bream (*Megalobrama terminalis* Richardson, 1846), barbel chub (*Squaliobarbus curriculus* Richardson, 1846), and common sawbelly (*Hemiculter leucisculus* Richardson, 1846) [50]. The larvae and young of these fish are an important food source of the piscivores.

Normally, in an undisturbed fish community with a stable food web, energy transfer goes from species that feed on zooplankton and invertebrates (i.e., lower δ^15^N values) to piscivore species (i.e., higher δ^15^N values), showing a continuous enrichment of δ^15^N values (i.e., 3.4‰; [51]) between trophic levels. However, the introduction of non-native fish species, such as the introduction of invasive Nile tilapia, will modify the recipient food webs and disturb the normal energy fluxes, eventually destroying the stability of the ecosystem [21].

Our results also showed that the trophic status of native fish did not increase, but decreased (decreased δ^15^N range represents decreased trophic levels or food chain length) in the invaded Dongjiang River, although Nile tilapia invasion forced some native fish to increase their food sources. Specifically, Nile tilapia invasion reduced the trophic status of native piscivores, invertivores, and omnivores (with a shorter δ^15^N range than the reference Beijiang River). This indicated that the food chain of these higher trophic status fish had been shortened. Species with different trophic positions have different impacts on the trophic structure of recipient communities. Non-native species with a high trophic position (such as top predators) will increase the length of the food chain through the addition mechanism [8]. Nile tilapia are omnivores and have a wide range of feeding habits, they can compete with any type of native fish for food, and can impact any part of the food chain, ultimately shortening food chains [21,47], resulting in native fish having to find new food resources, increasing the δ^13^C range [52]. Food chain stability is a key component of food web stability and trophic structure in a community [53]. Nile tilapia invasion induced significant changes in food chains, thereby disrupting food webs and destabilizing trophic structure in the invaded Dongjiang River.

Correspondingly, we also found that Nile tilapia invasion decreased the TA size of native species. This demonstrated isotopic niche compression of native species caused by a diet shift induced by competition with Nile tilapia. Specifically, Nile tilapia invasion compressed the isotopic niche space of native invertivores, omnivores, piscivores, and planktivores (all had a lower TA and a lower SEA than in the reference river). Trophic changes due to fish invasion can also exhibit biotic homogenization with trophic downgrading [54]. The size of the SEA of native fish communities in the invaded Dongjiang River was also significantly smaller than that in the reference Beijiang River, indicating that Nile tilapia were mainly located at the center of the isotopic niche space [8].

Generally, non-native species will increase the isotopic niche of the community through species addition, due to the functional and ecological differences between non-native species and native species [8,22]. Simultaneously, according to the principle of competitive exclusion, no two species have completely coincident niches in a community [47]. By adding species, a novel isotopic space will be added to the recipient communities, and isotopic niche size will increase. However, the isotopic niche size decreased instead of increased with the addition of Nile tilapia in the invaded Dongjiang communities. This indicated that Nile tilapia exerted serious and intense food competition on native species, resulting in the compression of the trophic space of native species in recipient communities.

Our research proved that the presence of Nile tilapia decreased resource availability for species at higher trophic levels, decreasing the maximal trophic position and food chain length in recipient communities. Specifically, Nile tilapia invasion shortened the food chain through the competition mechanism and modified the isotopic niche of native species [21]. There is a substantial overlap in diet between Nile tilapia and native fishes in most tropical and subtropical habitats [41]. In the downstream sections of the Pearl River, studies have shown that the invasion of Nile tilapia will decrease the body size of native economically important fish, impacting fish plumpness, body length, and body weight [50]. This also indirectly proves that there is strong competition for food between Nile tilapia and native fish species.

More importantly, the Nile tilapia invasion also has an impact on the isotopic diversity of the Dongjiang River. The IRic in the invaded Dongjiang River was significantly lower than that in the reference Beijiang River. This is consistent with the above results, that is, Nile tilapia competed with native species for food resources and resulted in the compression of the trophic niche size of local species, decreasing the total area of the trophic niche of the community. This also demonstrated that Nile tilapia had a high trophic niche overlap with native species [55]. The IEve in the invaded Dongjiang River was also significantly lower than that in the reference Beijiang River. This indicated that the abundance distribution among all fish in the stable isotope space was extremely uneven in the invaded community. Nile tilapia invasion disrupted the diet balance among fish in the recipient community, increased harassment in the isotopic space, and even caused resource competition among native species in the community. The invaded Dongjiang River had a higher IUni compared to the reference Beijiang River, suggesting higher isotopic redundancy in the fish community of the invaded Dongjiang River as a result of Nile tilapia invasion making more species compete for the same food resource.

Isotopic diversity can be used to track the multiple aspects of resource use of fish species in freshwater ecosystems and is considered to be an important variable in the structural stability of food webs [22]. Our analysis of isotopic diversity can be helpful in detecting the effects of invasive fish species on the trophic structure of the recipient community, and understanding subsequent impacts on ecosystem functioning. The invasion process of non-native species could restructure or even destroy food web structures, while changes in δ^13^C and δ^15^N values will be related to the diet shift of one or several species in a community, affecting the efficiency of energy transfer across trophic levels in food webs. The isotopic diversity indices used in the current study were a powerful supplement to assess the effects of non-native species on multiple facets of trophic structures and ecosystems.

Biological invasion is one of the leading causes of biodiversity loss, yet the mechanisms by which invaders progressively disrupt ecosystems is still a major issue in ecology. Over the past two decades, considerable attention has been focused on understanding the impact of invasive species on ecosystem functioning by analyzing the changes in nutritional interactions among species [56]. Although many studies have used stable isotope analyses to explore substantial ecological disturbances of non-native species and trophic destruction has been documented for a variety of taxa during the last two decades [57,58,59,60], this study provided clear evidence that invasive Nile tilapia can alter the food chain and influence trophic structure of native species, ultimately destroying ecosystem stability. In addition, Nile tilapia also compete with local fish for habitat and ultimately displace native fish from their preferred habitats [61]. The presence of Nile tilapia may cause a series of environmental problems, such as decreased water quality through sediment re-suspension (bioturbation), nutrient excretion [62]), and increasing nitrogen and phosphorus availability, promoting fast-growing algae [50].

The negative impact of Nile tilapia invasion on the aquatic ecosystem is currently attracting extensive attention, following its widespread introduction over the past 60 years [63]. Nile tilapia have been officially listed as one of the world’s top 100 invasive species in China from 2014. Despite this, the number of wild populations and invasions are increasing. This is mainly because Nile tilapia are one of the most important aquaculture species in the world and plays a very important role in the global freshwater fish trade [32]. Nile tilapia are also one of the most competitive aquaculture species in China [64]. The production of Nile tilapia in China reached 1.66 million tons in 2020, accounting for about 6% of all aquatic fisheries products in China [65]. For many years, more than 60% of global Nile tilapia exports have come from China [64]. The significant demand for Nile tilapia in the international market, makes it very difficult to control its invasion, and it is neither realistic nor desirable to completely eliminate it from China.

In fact, once a non-native species has successfully invaded, it is almost impossible to remove them. In order to better develop the Nile tilapia aquaculture industry, the breeding scale is expanding and germplasm is improving in China. Hybrid and mixed hybrid breeding will lead to a stronger intrusion capability. In view of its important economic and industrial status, its introduction cannot be banned. The prevention of Nile tilapia invasion is still a difficult problem. However, the serious impact of Nile tilapia on the trophic structure of native fish populations and ecosystems in southern China should not be ignored or underestimated. Therefore, the strictest supervision of this species is needed. At present, the most effective way to prevent the invasion of Nile tilapia in south China may be the implementation of strict isolation measures in pond culture, such as adding isolation fences beside the pond to prevent Nile tilapia escaping from the farm during floods. At the same time, special laws and regulations on the release of Nile tilapia as well as a scientific evaluation and monitoring system must be developed.

## 5. Conclusions

Our research provided empirical evidence that the invasion of Nile tilapia destroyed recipient ecosystem stability by forcing the low trophic status native herbivores and planktivores to seek new food sources, shortening food chains by decreasing the trophic levels of native invertivores, omnivores, and piscivores, and affected the isotopic diversity of native communities by compressing the isotopic niche space of native invertivores, omnivores, piscivores, and planktivores. The serious impact of Nile tilapia on the trophic structure of native fish populations and ecosystems in southern China should not be ignored despite its important economic and industrial status. Therefore, the strictest supervision of this species is needed.

## Figures and Tables

**Figure 1 biology-11-01665-f001:**
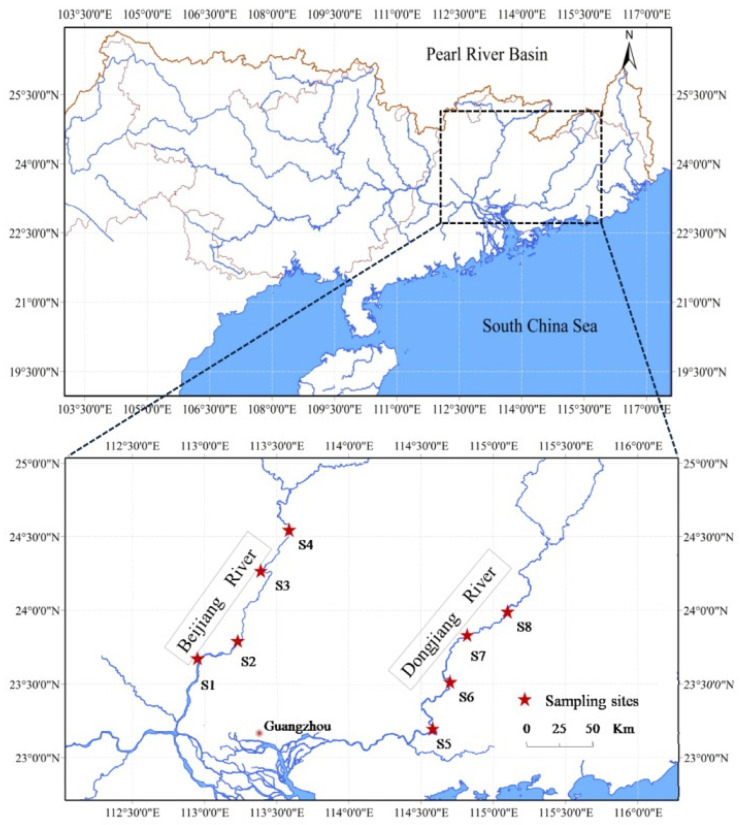
Location of sampling sites.

**Figure 2 biology-11-01665-f002:**
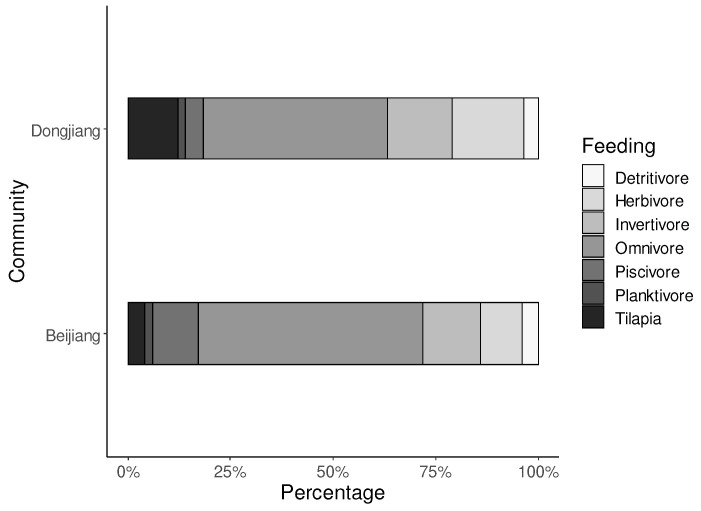
Quantity proportions of different feeding habits in the invaded Dongjiang River and the reference Beijiang River.

**Figure 3 biology-11-01665-f003:**
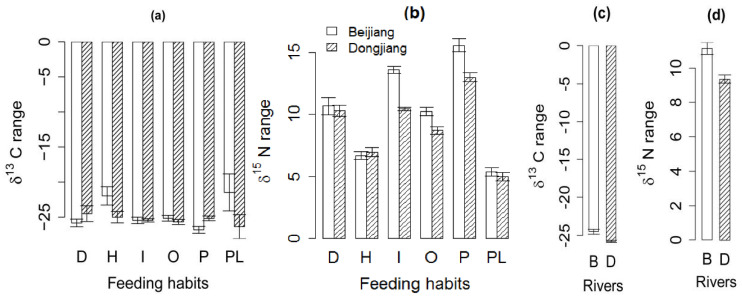
The difference in δ^13^C range and δ^15^N range between the invaded Dongjiang River and reference Beijiang River. B, Beijiang River; D, Dongjiang River. (**a**) Difference in δ^13^C range based on fish feeding habit. (**b**) Difference in δ^15^N range based on fish feeding habit. (**c**) Overall difference in δ^13^C range. (**d**) Overall difference in δ^15^N range.

**Figure 4 biology-11-01665-f004:**
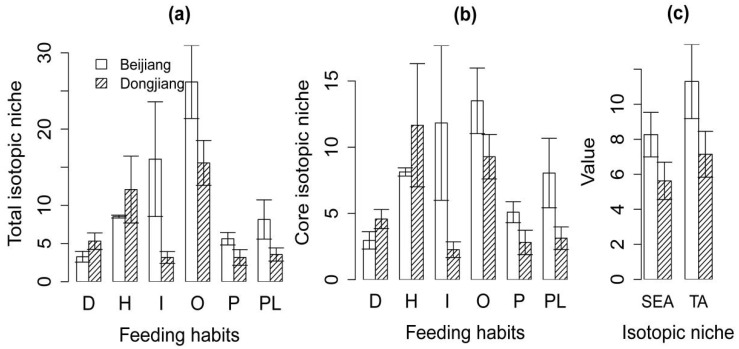
Differences in TA and SEA between the invaded Dongjiang River and the reference Beijiang River. (**a**) Difference in TA based on fish feeding habit. (**b**) Difference in SEA based on fish feeding habit. (**c**) Overall difference in TA and SEA.

**Figure 5 biology-11-01665-f005:**
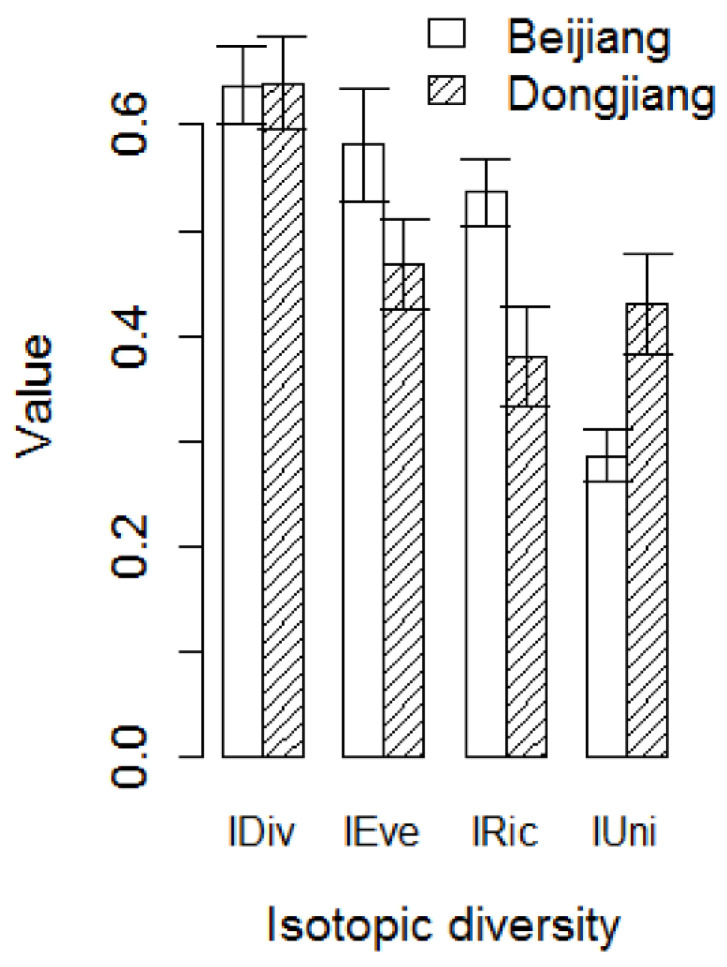
Differences in isotopic diversity index in the invaded Dongjiang River and the reference Beijiang River. IDiv, isotopic divergence; IRic, isotopic richness; IEve, isotopic evenness; IUni, isotopic uniqueness.

**Table 1 biology-11-01665-t001:** Basic information of the sampling sites in the rivers.

Sites	Name	Coordinates	River Width (m)	Species Richness	Subordinate River
S1	Lubao	112°53′23″ E, 23°20′53″ N	791	35	Beijiang
S2	Shijiao	112°57′59″ E, 23°33′41″ N	882	53	
S3	Qingyuan	113°3′49″ E, 23°41′50″ N	935	46	
S4	Lianjiang	113°18′16″ E, 24°1′29″ N	635	32	
S5	Hengli	114°36′55″ E, 23°10′26″ N	770	42	Dongjiang
S6	Guzhu	114°41′26″ E, 23°30′25″ N	462	50	
S7	Heyuan	114°42′45″ E, 23°44′18″ N	714	44	
S8	Huangtian	114°59′36″ E, 23°53′17″ N	341	21	

**Table 2 biology-11-01665-t002:** Fish community structure in the Dongjiang River and Beijiang River.

Feeding Habit	Species	English Name	Percentage (%)		Type	Category
			Beijiang	Dongjiang		
Piscivore	*Culter recurviceps*	Culter hainan	3.62	0.72	E	SE
	*Culter dabryi*	Dashi culter	1.04	−	N	SE
	*Culter alburnus*	Topmouth culter	0.07	0.59	N	SE
	*Erythroculter hypselonotus*	Bigeyse culterfish	0.07	+	N	SE
	*Elopichthys bambusa*	Yellow cheek carp	0.03	−	N	RL
	*Pelteobagrus fulvidraco*	Yellow catfish	1.45	0.71	N	SE
	*Pelteobagrus vachelli*	Darkbarbel catfish	1.32	1.49	N	SE
	*Leiocassis crassilabris*	Ussuri catfish	1.05	0.01	N	SE
	*Mystus guttatus*	Spotted longbarbel catfish	0.47	0.25	N	SE
	*Silurus asotus*	Catfish	0.31	0.22	N	SE
	*Clarias fuscus*	Oriental catfish	0.08	0.66	N	SE
	*Mystus macropterus*	Largefin longbarbel catfish	0.01	−	N	SE
	*Siniperca kneri*	Bigeye mandarinfish	0.36	0.07	N	SE
	*Siniperca scherzeri*	Spotted mandarinfish	0.15	−	N	SE
	*Channa asiatica*	Chinese snakehead	0.03	0.29	N	SE
	*Channa maculata*	Taiwan snakehead	0.01	0.17	N	SE
	*Channa argus*	Snakehead	0.01	−	N	SE
	*Anguilla japonica*	Japanese eel	+	0.02	N	RS
Invertivore	*Squalidus wolterstorffi*	Dot chub	3.14	0.05	N	RL
	*Saurogobio dabryi*	Longnose gudgeon	1.52	4.63	N	RL
	*Hemibarbus labeo*		1.22	0.77	N	SE
	*Hemibarbus maculatus*		1.20	1.13	N	SE
	*Opsariichthys bidens Günther*	Chinese hooksnout carp	1.00	0.90	N	SE
	*Coilia grayii*	Gray’s grsnadier anchovy	3.79	4.31	N	SE
	*Lateolabrax japonicus*	Spotted sea bass	0.67	+	N	RS
	*Eleotris oxycephala*	Sharphead sleeper	0.49	0.20	N	SE
	*Mastacembelus armatus*	Tire track eel	0.37	0.61	N	SE
	*Rhinogobius giurinus*	Amur goby	0.27	1.69	N	SE
	*Glossogobius giuris*	Tongue goby	−	2.79	N	SE
	*Hypseleotris hainanensis*		−	0.01	N	SE
	*Leiocassis argentivittatus*	Longitudinal catfish	0.24	0.33	N	SE
	*Ietalurus punetaus*	Channel catfish	−	0.07	Non	SE
	*Leiocassis virgatus*	Striped catfish	−	0.39	N	SE
	*Glyptothorax fukiensis*		−	0.09	N	SE
	*Monopterus albus*	Finless eel	0.07	0.05	N	RS
	*Takifugu ocellatus*	Ocellated puffer	+	−	N	RS
Omnivore	*Squalidus argentatus*	Chub	18.17	7.25	N	RL
	*Hemiculter leucisculus*	Common sawbelly	15.32	17.22	N	SE
	*Pseudohemiculter dispar*		3.54	0.36	N	SE
	*Squaliobarbus curriculus*	Barbel chub	2.92	1.50	N	RL
	*Abbottina rivularis*	Amur false gudgeon	2.60	0.02	N	SE
	*Cyprinus carpio*	Carp	1.94	1.52	N	SE
	*Carassius auratus*	Crucian	1.80	2.52	N	SE
	*Cirrhinus mrigala*	Mrigal carp	1.32	1.10	Non	SE
	*Sarcocheilichthys parvus*		0.90	−	N	SE
	*Rhodeus sinensis*	Light’s bitterling	0.57	−	N	SE
	*Xenocypris davidi*	Yellow tailed xenocypris	0.36	2.29	N	RL
	*Hemiculterella wui*		0.22	−	E	SE
	*Puntius semifasciolatus*	Chinese barb	0.24	−	N	SE
	*Mylopharyngodon piceus*	Black carp	0.01	0.01	N	RL
	*Osteochilus salsburyi*		0.11	1.01	N	SE
	*Xenocypris argentea*	Silver xenocypris	0.12	0.05	N	RL
	*Distoechodon tumirostris*	Round mouth	0.05	0.02	N	RL
	*Acheilognathus tonkinensis*	Vietnamese bitterling	0.03	0.61	N	SE
	*Acheilognathus macropterus*	Largefin bitterling	0.02	−	N	SE
	*Cyprinus carpio* var.*specularis*	Germany mirror carp	0.01	−	N	SE
	*Acheilognathus chankaensis*	Khanka spiny bitterling	−	0.26	N	
	*Sarcocheilichthys nigripinnis*		−	0.15	N	
	*Pseudorasbora parva*	Stone moroko	−	0.02	N	SE
	*Spinibarbus denticulatus*		−	0.02	N	RL
	*Rhodeus spinalis Oshima*		−	0.01	N	SE
	*Zacco platypus*	Pale chub	3.40	0.09	N	SE
	*Tinca tinca*	Tench	−	0.01	Non	SE
	*Oreochromis niloticus*	Nile tilapia	3.65	13.21	Non	SE
	*Tilapia zillii*	Zillii tilapia	0.27	0.70	Non	SE
	*Anabas testudineus*	Climbing perch	−	0.01	Non	SE
	*Prochilodus scyofa*		0.17	0.01	Non	SE
	*Clarias gariepinus*	Fuscous catfish	0.01	0.18	Non	SE
Detritivore	*Misgurnus anguillicaudatus*	Oriental weather fish	4.03	3.50	N	SE
	*Micronoemacheilus pulcher*		0.21	0.08	E	SE
	*Cobitis sinensis*	Siberian spiny loach	0.01	0.38	N	SE
	*Labeo rohita*	Roho labeo	0.10	−	Non	SE
	*Vanmanenia hainanensis*		−	0.01	E	SE
	*Hypostomus plecostomus*	Suckermouth catfish	0.05	0.05	Non	SE
Herbivore	*Cirrhinus molitorella*	Mud carp	5.67	12.55	N	RL
	*Megalobrama terminalis*	Black amur bream	2.72	6.74	N	RL
	*Ctenopharyngodon idellus*	Grass carp	0.80	0.81	N	RL
	*Sinibrama wui*	Bigeyes bream	0.40	0.09	E	RL
	*Onychostoma gerlachi*	Largescale shoveljaw fish	0.31	−	N	SE
	*Acrossocheilus beijiangensis*		0.13	−	N	SE
	*Parabramis pekinensis*	White bream	0.10	0.06	N	RL
	*Megalobrama amblycephala*	Wuchang fish	0.05	0.02	Non	RL
	*Acrossocheilus labiatus*		0.03	−	N	SE
	*Acrossocheilus stenotaeniatus*		0.02	−	N	SE
	*Acrossocheilus parallens*		0.02	−	N	SE
	*Sinibrama melroseib*	Hainan bream	0.02	0.06	N	SE
	*Rectoris posehensis*		0.01		N	SE
	*Garra orientalis*	Oriental sucking barb	−	0.04	N	SE
	*Parasinilabeo assimilis*		−	0.01	N	SE
Planktivore	*Hypophthalmichthys molitrix*	Silver carp	1.16	1.52	N	RL
	*Aristichthys nobilis*	Bighead carp	0.77	0.39	N	RL
	*Pseudolaubuca sinensis*		−	0.03	N	SE
	*Clupanodon thrissa*	Chinese gizzard shad	0.19	−	N	RS
	*Konosirus punctatus*	Dotted gizzard shad	0.05	−	N	RS

E, endemic to China; N, native species; Non, non-native species; RS, river-sea migratory; RL, river-lake migratory; SE, sedentary; “+” indicates occasional species, feeding habit, type, and category of fish were determined according to Zheng [45] and Zhou and Zhang [46].

**Table 3 biology-11-01665-t003:** Effects of Nile tilapia and native species of different trophic levels on δ^15^N range, δ^13^C range, and the total isotopic niche of native species in the invaded Dongjiang River and the reference Beijiang River tested using mixed effect models.

River	Variable	Intercept	No. of Native PL.	No. of Native H	No. of Native D	No. of Native O	No. of Native I	No. of Native P	No. of Nile Tilapia
Dongjiang	δ13C range	1.45 (**0.039**)	1.06 (0.068)	−0.02 (0.594)	0.04 (0.261)	0.10 (0.075)	0.05 (0.714)	0.18 (**0.019**)	0.22 (**0.030**)
	δ15N range	1.52 (**0.026**)	1.55 (**0.031**)	0.03 (0.298)	0.05 (0.158)	0.19 (0.064)	0.02 (0.853)	0.25 (**0.008**)	−0.19 (**0.025**)
Beijiang	δ13C range	1.25 (**0.018**)	0.14 (0.332)	−0.01 (0.381)	0.23 (**0.042**)	−0.01 (0.491)	0.19 (0.168)	0.22 (**0.037**)	0.04 (0.453>)
	δ15N range	1.32 (0.025)	0.11 (0.216)	0.05 (0.631)	0.19 (0.712)	0.07 (0.329)	0.12 (0.078)	0.24 (**0.036**)	0.08 (0.371)

Significant *p*-values < 0.05 are displayed in bold. (P: piscivores, I: invertivores, O: omnivores, D: detritivores, H: herbivores and PL.: planktivores).

## Data Availability

The datasets used and/or analyzed during the current study are available from the corresponding author on reasonable request.

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
