# Peer review of "Nile Tilapia (Oreochromis niloticus Linnaeus, 1758) Invasion Caused Trophic Structure Disruptions of Fish Communities in the South China River—Pearl River"

_biology, 2022, doi:10.3390/biology11111665_

Round 1

Reviewer 1 Report

In this study, the authors studied the effect of Nile tilapia invasion on the trophic structure of fish communities based on two rivers, the one has a serious Nile tilapia invasion and the other one has a relatively small Nile tilapia population, which can help us to understand the mechanism of invasive species to disrupt the food web and trophic structure of native species. It’s a very interesting study. However, there are somewhere should be improved in the manuscript.

1. In the introduction, the contents of the 5th and 6th paragraphs are similar, these two paragraph are better to merge into one paragraph.

2. Line 71-73, the invasion of lake trout forces the grizzly bears and black bears to seek alternative food resources, which is lack of reference.

3. Figure 1 should be improved, which is lack of compass, the scale of map, etc. It was failure to show where these two rivers located in China.

4. Table 1, the header of the table can be improved, e.g, width -- river width,; species number – species richness. And the legend of table can be improved.

5. Line 135-138, the sampling methods should be described in detail. How many days did the sample per month? How long did they sample per time for different fishing nets ?

5. The statistical analyses part should be improved in details. E.g, what kind of data did the authors used for linear mixed effect model ? Did they use the raw data or data transformed data ? Were the data in normal distribution ?

6. Fig.2, the percentage based on biomass or abundance?

7. Table 2, why did the authors list fish list like this ? The fish list should be listed in a certain rules. By the way, the authors mentioned rare species. So how did the authors categorize the rare species ?

8. Figure 3, This is a figure showing that the difference in δ13C range and δ15N range between two rivers. I want to know how did the author calculate the δ13C range and δ15N range in two different rivers. Did the data contain another non- native species ?

9. Moderate English changes required.

Line13, show—showed;

Line 22 herbivorous and planktivorous fish – herbivores and planktivores;

The text font in the full manuscript was too messy.

Author Response

Responses to Reviewers' Comments

(For the ease of response, we number the comments)

Dear Reviewer 1,

We would like to express our sincere thanks to your constructive comments and suggestions. Based on your comments and suggestions, we have revised the manuscript carefully and deeply. We reply to all the comments point-by-point below (bold).

===========

 Q1

In the introduction, the contents of the 5th and 6th paragraphs are similar, these two paragraph are better to merge into one paragraph.

Response: Thanks for the suggestion. We have merged the 5th and 6th paragraphs into one paragraph now.

Q2

Line 71-73, the invasion of lake trout forces the grizzly bears and black bears to seek alternative food resources, which is lack of reference.

Response: Thanks for the suggestion. We have added the reference into that sentence now.

 Q3

Figure 1 should be improved, which is lack of compass, the scale of map, etc. It was failure to show where these two rivers located in China.

Response: Thanks for the suggestion. We have improved Figure 1, added compass, scale and its location in China now.

Q4

Table 1, the header of the table can be improved, e.g, width -- river width; species number – species richness. And the legend of table can be improved.

Response: Thanks for the suggestion. We have changed "width" into "river width", changed "species number" into "species richness", and revised the legend into "Basic information of the sampling sites in the rivers" now.

Q5

Line 135-138, the sampling methods should be described in detail. How many days did the sample per month? How long did they sample per time for different fishing nets ?

Response: Thanks for the suggestion. We have added the following information into the Data Collection section "Catch samples were collected over one to three successive days at each site, according to favorable weather conditions. ", "Sampling was performed using the same protocol in each year. Sampling started in the afternoon (approx. 18:00 pm) and last 12 hours for a whole night for all nets. " to describe the sampling methods more detailed.

Q6

The statistical analyses part should be improved in details. E.g, what kind of data did the authors used for linear mixed effect model ? Did they use the raw data or data transformed data ? Were the data in normal distribution ?

Response: In the LM analysis, we used the lg(x+1) transformed data. We add this sentence "The abundance data were lg(x+1) transformed before analysis to conform to the normal distribution." into the Statistical Analyses section to make it more clearly now.

Q7

Fig.2, the percentage based on biomass or abundance?

Response: the percentage based on abundance. We have changed the figure legend into "Quantity proportions of different feeding habits in the invaded Dongjiang River and the reference Beijiang River." to make it more clearly now.

Q8

Table 2, why did the authors list fish list like this ? The fish list should be listed in a certain rules. By the way, the authors mentioned rare species. So how did the authors categorize the rare species ?

Response: In this study, We divided native species into six categories (piscivore, invertivore, omnivore, detritivore, herbivore, and planktivore) based on their feeding habits, in order to analyze the effects of Nile tilapia invasion on different fish trophic niches. So in table2, we listed the species according to their feeding habits, in order to distinguish the proportion of different trophic categories in the community more directly. We have added the order and family information of each species into Table 2 to make it more scientific now.

We means occasional species. We have used "occasional species" instead "rare species" now.

Q9

Figure 3, This is a figure showing that the difference in δ13C range and δ15N range between two rivers. I want to know how did the author calculate the δ13C range and δ15N range in two different rivers. Did the data contain another non- native species ?

Response: The δ13C range and δ15N range of native species in each river were calculated as the differences between maximal and minimal values for δ13C and δ15N to compare the difference in resource use and food chain length between the invaded and reference rivers. The data does not contain another non-native species. We described these in Stable Isotope Metrics section.

Q10

Moderate English changes required.

Response: Thanks for the suggestion. We have polished the English again now.

Q11

Line13, show—showed;

Response: Thanks for the suggestion. We have change "show" into "showed" now.

Q12

Line 22 herbivorous and planktivorous fish – herbivores and planktivores;

Response: Thanks for the suggestion. We have change "herbivorous fish", "piscivorous fish", "invertivorous fish", "omnivorous fish", "detritivorous fish", "planktivorous fish" into "herbivores", "piscivores", "invertivores", "omnivores", "detritivores", and "planktivores" respectively in the whole text now.

Q13

The text font in the full manuscript was too messy.

Response: We apologize for this messy text font. We have modified the font and adopted the uniform font in the full manuscript now.

Reviewer 2 Report

Dear Author,

Necessary corrections are indicated in the text as stick notes. 

Author Response

Responses to Reviewers' Comments

(For the ease of response, we number the comments)

Dear Reviewer 2,

We would like to express our sincere thanks to your constructive comments and suggestions. Based on your comments and suggestions, we have revised the manuscript carefully and deeply. We reply to all the comments point-by-point below (bold).

===========

 Q1

(Linnaeus, 1758)

Should be given name of the author who first defined the species.

Response: Thanks for the suggestion. We have added the given name of the author who first defined this species into the species name now.

Q2 

keywords must be different from the title.

Response: Thanks for the suggestion. The keywords have been changed into "invasion; isotope structure; trophic status; isotopic diversity; isotopic niche" now.

Q3

"For instance, the establishment of Round Goby (Neogobius melanostomus) has been 38 associated with decreases in the occurrence of native fish such as Mottled Sculpin (Cottus bairdii) in the Laurentian Great Lakes [6], three-spined Stickle-back (Gasterosteus aculeatus) 40 in the Gulf of Gdansk, and the protected River Bullhead (Cottus perifretum) in the River 41 Meuse, Netherlands."

Should be given name of the author who first defined the species.

Response: Thanks for the suggestion. We have added the given name of the author who first defined this species into the species name in the whole text now.

Q4

Figure 1 Add a compass and scale to the map, if possible, provide a better map.

Response: Thanks for the suggestion. Combined with the comments of reviewer 1 on question 3, we have improved Figure 1, added compass, scale and its location in China.

Q5 

Table 2.  Add "-" sign for all values not given in the table. And some species should be given left-justified.

Response: Thanks for the suggestion. We have added "-" sign for all values not given in the table and left-justified some species now.

Q6 

'herbivorous fish'. spelling should be checked

Response: Thanks for the suggestion. Combined with the comments of reviewer 1 on question 12, We have change "herbivorous fish", "piscivorous fish", "invertivorous fish", "omnivorous fish", "detritivorous fish", "planktivorous fish" into "herbivores", "piscivores", "invertivores", "omnivores", "detritivores", and "planktivores" respectively in the whole text now.

Reviewer 3 Report

Biology1987637

Major comment

1.    Need to check the font throughout the text, since larger and smaller fonts are used in several places, such as lines 22, 23, 24, 25, 60, 208-218, 219-224, 225-234, 236-239, 269-275, 280-283, 297-298

  Minor comments

1.    Line 66: add space after organisms

2.    Line 148: Massachusetts should be MO

3.    Line 149: delete Waltham, Massachusetts, USA

4.    Figure 5. In the square beijiang and dongjiang should be Beijiang and Dongjiang

5.    Line 431: scientific name of animals should be italic

6.    Line 445: mediterranean should be Mediterranean

7.    Line 465: scientific name of animals should be italic

8.    Line 496-497: title should be small capital

9.    Line 507. Biochem. should be italic

10. Lines 510-511: ciprinus  oreochromis should be Ciprinus  Oreochromis (italic)

11. Lines 513-514: title should be small capital

12. Line 521: add city and country after Press

13. Line 532: is this a book? If so, book title should be large capital for the first letter of each word, add publisher, city and country

14. Line 540-541: scientific names of animals should be italic

15. Line 561: journal name should be italic

16. Line 563: book title should be large capital for the first letter of each word

Author Response

Responses to Reviewers' Comments

(For the ease of response, we number the comments)

Dear Reviewer 3,

We would like to express our sincere thanks to your constructive comments and suggestions. Based on your comments and suggestions, we have revised the manuscript carefully and deeply. We reply to all the comments point-by-point below (bold).

===========

 Q1

Need to check the font throughout the text, since larger and smaller fonts are used in several places, such as lines 22, 23, 24, 25, 60, 208-218, 219-224, 225-234, 236-239, 269-275, 280-283, 297-298.

Response: Thanks for the suggestion. We apologize for this messy text font. We have modified the font and adopted the uniform font in the full manuscript now.

Q2

  Line 66: add space after organisms

Response: Thanks for the suggestion. We add space after organisms now.

Q3

   Line 148: Massachusetts should be MO

Response: Thanks for the suggestion. We have changed "Massachusetts" into "MO" now.

Q4

   Line 149: delete Waltham, Massachusetts, USA

Response: Thanks for the suggestion. We have deleted "Waltham, Massachusetts, USA" from line 149 now.

Q5

Figure 5. In the square beijiang and dongjiang should be Beijiang and Dongjiang.

Response: We apologize for this mistake. We have modified "beijiang" and "dongjiang" into "Beijiang" and "Dongjiang" in Figure 5 respectively now.

Q6

  Line 431: scientific name of animals should be italic.

Response: We apologize for this mistake. We have modified the scientific name Neogobius melanostomus into italic now.

Q7

Line 445: mediterranean should be Mediterranean.

Response: We apologize for this mistake. We have modified "mediterranean" into "Mediterranean" now.

Q8

   Line 465: scientific name of animals should be italic.

Response: We apologize for this mistake. We have modified the scientific name "Salvelinus fontinalis" and "Salmo trutta" into italic now.

Q9

  Line 496-497: title should be small capital.

Response: We apologize for this mistake. We have changed the title into "Tradeoffs among ecosystem services associated with global tilapia introductions" now.

Q10

Line 507. Biochem. should be italic.

Response: We apologize for this mistake. We have changed "Biochem" into italic now.

Q11

Lines 510-511: ciprinus oreochromis should be Ciprinus Oreochromis (italic). 

Response: We apologize for this mistake. We have changed "ciprinus oreochromis" into italic "Ciprinus Oreochromis", and changed "oreochromis niloticus" into "Oreochromis niloticus" now.

Q12

Lines 513-514: title should be small capital.

Response: We apologize for this mistake. We have changed the title into "Assessing the influence of tilapia on sport fish species in north Carolina reservoirs" now.

Q13

Line 521: add city and country after Press

Response: We have add the city "Beijinga" and country "China" after Press now.

Q14

Line 532: is this a book? If so, book title should be large capital for the first letter of each word, add publisher, city and country.

Response: It is a journal. We have now revised the format.

Q15

Line 540-541: scientific names of animals should be italic.

Response: We apologize for this mistake. We have changed "Ambystoma mexicanum", "Cyprinus carpio" and "Oreochromis niloticus" into italic "Ambystoma mexicanum", "Cyprinus carpio" and "Oreochromis niloticus" now.

Q16

Line 561: journal name should be italic.

Response: We apologize for this mistake. We have changed "Biol Invasions" into "Biol Invasions" now.

Q17

Line 563: book title should be large capital for the first letter of each word.

Response: We apologize for this mistake. We have changed the book title into "Technology of Ecological and Efficient Culture of Tilapia" now.

Round 2

Reviewer 1 Report

Dear editors, 

The authors of this manuscript have already revised the whole manuscript based on the comments. It can be accepted after minor revison.

Kindest Regards

Lianglilang Huang

Author Response

Dear Reviewer 1,

We would like to express our sincere thanks to you and reviewers for your constructive comments and suggestions. Based on your comments and suggestions, we have revised the manuscript carefully and deeply again. The main modifications are as follows:

(1) Modified the figures, such as changed "community" into "Community" of Y-axis in Figure 2, superscripted the number of the Y- axis in Figure 3, deleted two same legends from Figure 4, changed "value" into "Value" of Y-axis in Figure4c and Figure 5.

(2) Checked the references and deleted two irrelevant to the manuscript and rearranged the references.